# FFC-NMR Power Supply with Hybrid Control of the Semiconductor Devices

**António Roque** [1,*]**, Duarte M. Sousa** [2]**, Pedro J. Sebastião** [2]**, Vítor Silva** [3] **and Elmano Margato** [3]

1 Escola Superior de Tecnologia de Setúbal/IPS & INESC-ID, Campus do IPS, Estefanilha,
2910-761 Setúbal, Portugal
2 Instituto Superior Técnico, Universidade de Lisboa, Av. Rovisco Pais 1, 1049-001 Lisboa, Portugal;
duarte.sousa@tecnico.ulisboa.pt (D.M.S.); pedro.jose.sebastiao@tecnico.ulisboa.pt (P.J.S.)
3 Instituto Superior de Engenharia de Lisboa IPL, Rua Conselheiro Emídio Navarro 1,
1959-007 Lisboa, Portugal; vsilva@deetc.isel.ipl.pt (V.S.); efmargato@isel.ipl.pt (E.M.)
* Correspondence: antonio.roque@estsetubal.ips.pt

**Abstract:** The performance of FFC-NMR power supplies is evaluated not only considering the technique requirements but also comparing efficiencies and power consumption. Since the characteristics of FFC-NMR power supplies depend on the power circuit topology and on the control solutions, the control design is a core aspect for the development of new FFC systems. A new hybrid solution is described that allows controlling the power of semiconductors by switches (ON/OFF mode) or as a linear device. The approach avoids over-design of the power supply and makes it possible to implement new low power solutions constituting a novel design by joining a continuous match between the ON/OFF mode and the linear control of the power semiconductor devices.

**Keywords:** power supply; semiconductor devices; NMR; upward transient; downward transient; hybrid control

## 1. Introduction

A fast field cycling—nuclear magnetic resonance (FFC-NMR) apparatus is a set of several electronic circuits and magnetic devices (Figure 1) [1–7] from which the magnet and its power supply can be singled out as the core elements.

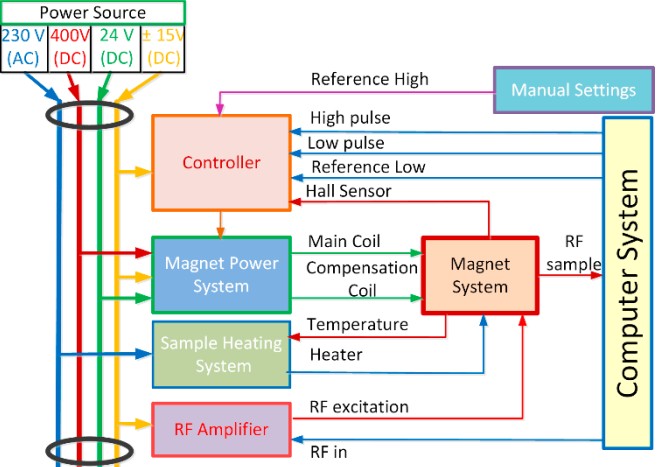

**Figure 1.** Diagram of the blocks and signals of the FFC-NMR apparatus.

FFC magnets are typically designed in line with the design type of the power supplies [8–14]. The most common FFC magnets are air-cored and use aluminum or copper as

conductive materials [9–11,15–18]. The magnet designs are the result of complex optimization algorithms and computational effort. Furthermore, their manufacture can become a considerable complex process that is both time consuming and costly.

In line with the power supply design described in this paper, a magnet with a ferromagnetic core was developed [19], which presents the following electrical parameters:

- Electrical resistance: $R_M = 3\ \Omega$;
- Self-inductance: $L_M = 270$ mH;
- Magnetic flux density/magnet current ratio: $\frac{B}{i_M} = 0.05$ T/A;
- Maximum magnet current: $i_{M\_max} = 5$ A.

The use of this magnet in FFC-NMR requires a power supply design with the following specifications [20–22]:

- Cycling the magnetic flux density between at least two different levels [1–3];
- Accurate unlimited repetition of the magnetic flux density cycles;
- Accurate current level definition (current error better than 0.01%);
- Current ripple: $\frac{\Delta i_M}{i_M} < 10^{-2}$;
- Linear transitions between magnetic flux density levels ("upward" and "downward");
- Fast "upward" and "downward" transients (less than 3.5 ms).

The equivalent circuit of the FFC-NMR power supply optimized according to the requirements above is showed in Figure 2. The main components of this circuit are [1,4,5]:

- Main power source, $U_0$;
- Auxiliary power source, $U_{aux}$;
- IGBT semiconductor, S (IGBT IXGH16N170);
- Auxiliary semiconductor, $S_{aux}$ (MOSFET SPP11N60S5);
- Magnet filter ($D_f$, $R_f$);
- Semiconductor snubber ($C_{sn}$, $R_{sn}$) and varistor (Var);
- Diode, D;
- Equalization resistor, $R_E$;
- Power supply main ON/OFF switch, S.

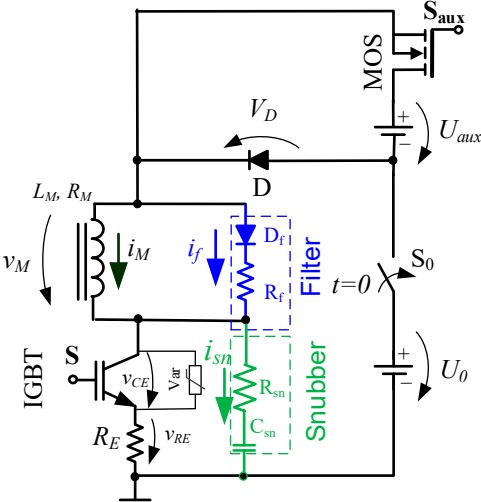

**Figure 2.** Equivalent electric circuit of the power supply.

The control solutions that will determine the operation of the semiconductors, S and $S_{aux}$, must consider the dynamic nature of the magnet current. The optimized solution proposed in this work is hybrid and considers an ON/OFF operation of semiconductors mixed with their operation in the active region [19,23–26]. Merging the non-conventional ON/OFF and linear control modes constitutes a technical novelty in the FFC-NMR field that led to a low power solution with a single-power semiconductor for a ferro-electromagnet.

## 2. Operation Modes Methods

The expected outcome of the FFC-NMR power supply is the precise control of the magnet current levels and commutations that must be fast and linear [27–29]. Considering the core current loop illustrated in Figure 3, these requirements are additionally constrained by the $\frac{dv}{dt}$ of the inductive loop and the expected low power consumption and increased efficiency of the power source.

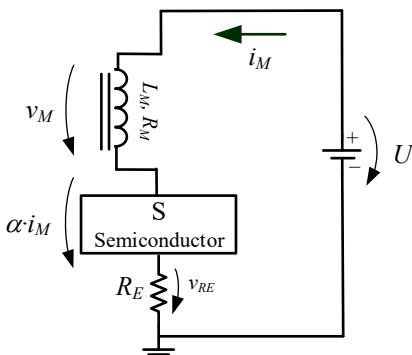

**Figure 3.** Basic inductive current loop of the FFC power supply.

Considering the Figure 3 circuit, the magnet current, and therefore the current transitions, can be changed if changing the voltage $U$ and the parameter $\alpha$.

Generically, $U$(t) is described by:

$$U = R_M i_M + L_M \frac{di_M}{dt} + \alpha i_M + R_E i_M \tag{1}$$

The conditions above require that the semiconductor S operates:

1. As an ON/OFF switch;
2. Linearly.

Therefore, the parameter $\alpha$ must be set according to each type of transient and the characteristics of the power semiconductor. Technically, this parameter can be estimated from the technical $v_{CE} = f(i_{CE})$ curves provided by the manufacturers or obtained experimentally by testing the power semiconductor.

### 2.1. Upward Transient

In order to perform efficiently, the required upward current transition, the voltage $U$, must be as high as possible and the semiconductor should behave as an ideal switch. Thus, for the upward transition, Figure 4 represents the magnet's current loop.

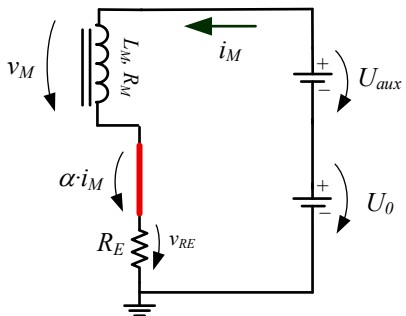

**Figure 4.** Basic inductive current loop for an "upward" magnet current transition.

In line with the Figure 2 circuit, $U = U_0 + U_{aux}$ and $\alpha = 0$.

Equation (1) becomes:

$$U_0 + U_{aux} = R_M i_M + L_M \frac{di_M}{dt} + R_E i_M \tag{2}$$

The time evolution of the magnet current is given by:

$$i_M(t) = \frac{U_0 + U_{aux}}{R_M + R_E} e^{-\frac{t}{\tau}}, \quad \tau = \frac{L_M}{R_M + R_E} \tag{3}$$

where $\tau$ is the decay time constant, which is mainly dependent on the magnet's resistance, $R_M$.

### 2.2. Downward Transient

To perform a downward current transient, it is not enough to open the semiconductor, S, since this action corresponds to the interruption of an inductive current loop originating a high $\frac{dv}{dt}$ rate, which can damage the semiconductors.

As a first approach, this problem can be avoided if the semiconductor, S, behaves as a capacitor, as represented in Figure 5 as $\alpha = \frac{v_S}{i_M}$.

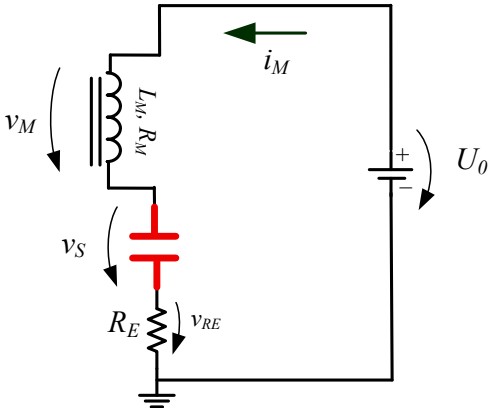

**Figure 5.** Downward equivalent electric circuit of the FFC power supply.

In this case, Equation (1) takes the form:

$$U_0 = R_M i_M + L_M \frac{di_M}{dt} + v_S + R_E i_M \tag{4}$$

During the downward transient $v_S = f(i_M)$, the magnet current steady state ($i_M$) is achieved when:

$$i_M = \frac{U_0 - v_S(I_M)}{R_M + R_E} \tag{5}$$

Since $v_S = f(i_M)$ is intrinsic to the semiconductor's characteristics (IGBT type as reference), this transient is in line with the dynamic response of a second-order electric circuit requiring the control of $v_S$ in order to fulfil the FFC-NMR requirements.

### 3. Hybrid Control

The magnet and the power supply developed under the FFC-NMR constraints are designed to obtain a linear relationship between the magnet current and the magnetic flux density in the probe room inside the magnet.

In Figure 6, the blocks and signals necessary to implement the described hybrid control solution are represented.

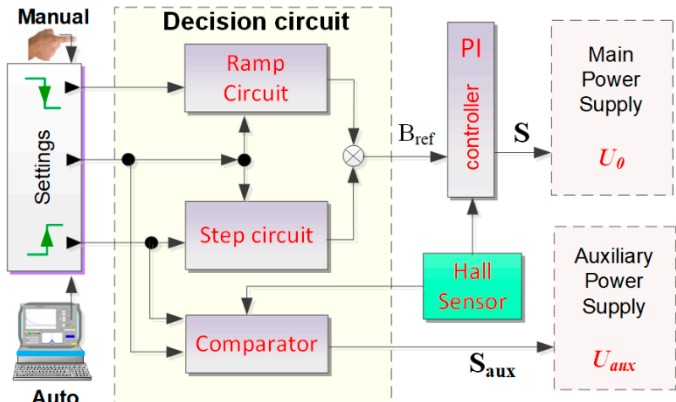

**Figure 6.** Control circuit of the developed FFC power supply.

As described in Section 2, the behavior of the semiconductor S is different for the "upward" and the "downward" transitions of the magnet current. Since the control feedback signal is obtained with a Hall effect sensor, which measures the magnetic flux density at the probe position, a central proportional–integral controller unit can be used (PI block in Figure 6).

For the "upward" transition of the magnet current, i.e., the magnet current transient from one magnetic flux density level to another higher level, the control system activates the state ON of the semiconductors S and $S_{aux}$ (see Figure 6). Thus, both semiconductors behave as switches. During this transient time, $t_{on}$, the command circuit activates the control loop corresponding to the "step circuit" block keeping the "ramp generator" control loop in stand-by (Figure 6). When the magnetic flux density reaches the upper-level target, the control circuit turns $S_{aux}$ OFF and keeps S ON. From that moment on, both the magnet current and $v_s$ remain constant. In this case, the steady-state equivalent circuit of the power supply is represented in Figure 7.

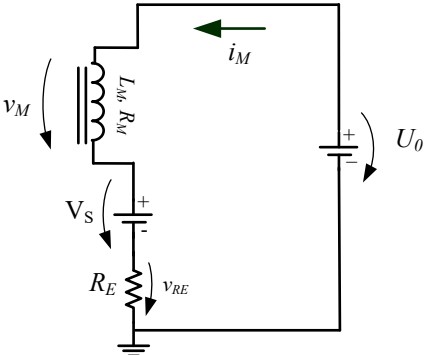

**Figure 7.** Steady-state equivalent electric circuit of the developed FFC power supply.

The magnet's steady-state current is given by

$$i_M = \frac{U_0 - V_S}{R_M + R_E} \tag{6}$$

During the "downward" transient, semiconductor S is controlled by a "ramp generator" circuit that imposes a linear decrease of the voltage $v_S$ with given slew rate. This slew rate takes into consideration the FFC technique specifications that requires the transients to be performed fast (typically less than 3.5 ms) provided that $\frac{dv}{dt}$ does not exceed the maximum value supported by the semiconductor, S.

## 4. Experimental Results

The proposed hybrid control solution was tested and the prototype is on operation.

The main experimental results correspond to the "upward" and "downward" magnet's current transitions, which are presented in Figure 8 together with the reference control signals for the magnetic field reference ($B_{ref}$). The signals were measured with ammeter and voltage probes connected to a digital oscilloscope. As it can be seen in Figure 8a the $B_{ref}$ is a step-up signal that switches S and $S_{aux}$ to the ON state. For a downward transition $B_{ref}$ is a ramp signal with a controlled slope for the desired maximum of 3.5 ms transition time. As it can be seen the dynamic behavior of the magnet current is linear in both cases.

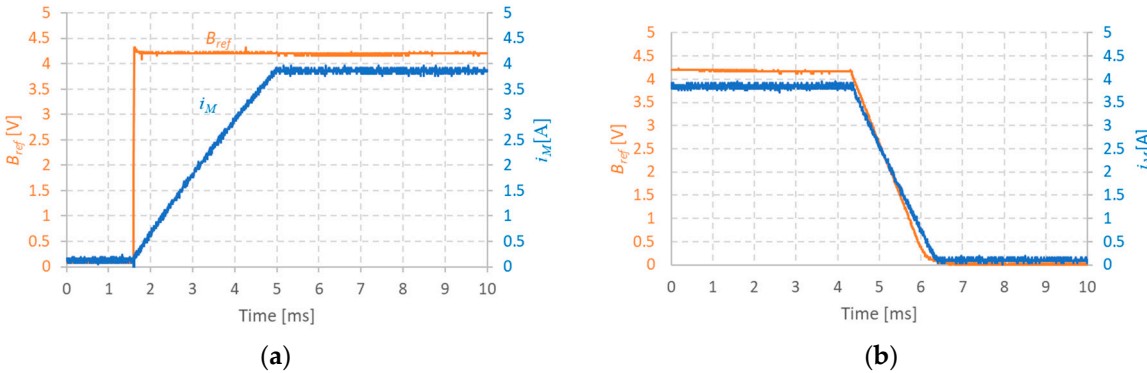

**Figure 8.** Time evolution of the magnetic field reference signal ($B_{ref}$) and magnet's current ($i_M$) for a magnetic field: (**a**) Upward transition; (**b**) Downward transition.

For comparison, the magnetic flux density (B) measured with the control system's Hall effect sensor is presented in Figure 9 together with $B_{ref}$. Using a calibrated Hall probe, it was observed experimentally, that for a $B_{ref}$ = 4.2, V corresponds a real B of 0.2 T and a magnet current $i_M$ of 3.8 A. Clearly, the dynamic behavior of *B*, in Figure 9, and $i_M$, in Figure 8, are very similar, from which a $\frac{B}{i_M}$ ratio of about 0.053 T/A is observed, in line with the ratio of 0.05 T/A initially specified.

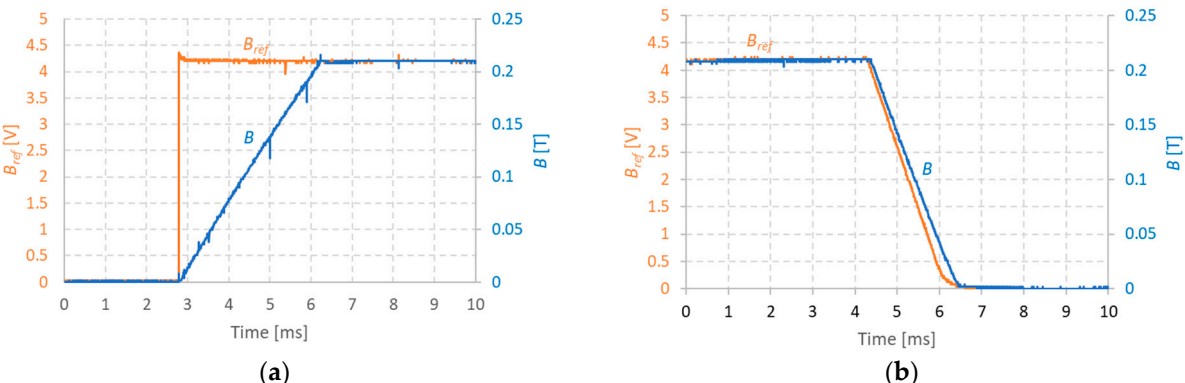

**Figure 9.** Experimental magnetic flux density transitions for a magnetic field: (**a**) Upward transition; (**b**) Downward transition.

It is important to note that both the upward and downward transient times observed in Figures 8 and 9 are considerably shorter (~3.5 ms) than the natural constant time ($\tau_n$) of the circuit, which for the prototype magnet is (see for example Equation (3)):

$$\tau_n \approx \frac{L_M}{R_M} \approx 90 \text{ ms} \tag{7}$$



For the upward transition (Figures 8a and 9a), the short transient time was possible because in view of Equation (2), which can be approximated by:

$$U_0 + U_{aux} \approx L_M \frac{\Delta i_M}{\Delta t} \tag{8}$$

for small $R_M$ and $R_E$ gives the possibility to adjust the transient time $\Delta t$ with help the auxiliary voltage $U_{aux}$.

For a nominal value of $B = 0.2$ T and for the ratio $\frac{B_M}{i_M} = 0.05$ T/A, the minimum value for $U_{aux}$ for a target $\Delta t = 3$ ms is:

$$(U_0 + U_{aux})_{min} \approx 360 \text{ V} \tag{9}$$

The experimental results shown in Figures 8a and 9a were obtained with $U_0 = 24$ V and $U_{aux} = 400$ V.

For the upward transition, the IGBT S gate-emitter voltage ($v_{GE}$) is in the saturation regime and $v_{GE}$ roughly constant as shown in Figure 10a.

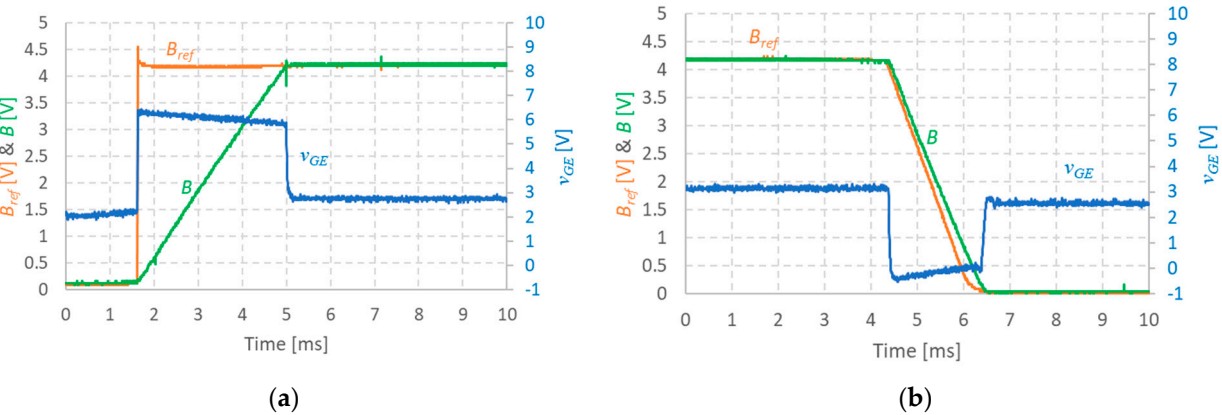

(**a**)　　　　　　　　　　　　　　　　(**b**)

**Figure 10.** Experimental IGBT gate voltage for a magnetic field: (**a**) Upward transition; (**b**) Downward transition.

In the case of the "downward" transition, the current decay's inductive nature (that would be characterized by $\tau_n$) was compensated with use of a direct control of the IGBT S gate's voltage. In fact, Equation (5) can be approximated by:

$$U_0 - v_s \approx L_M \frac{\Delta i_M}{\Delta t} \tag{10}$$

and, provided that $v_s$ is constant, $\frac{\Delta i_M}{\Delta t}$ is also approximately constant and a linear decay is obtained by precise control of $v_s$.

The S command voltage $v_{GE}$ necessary to keep $v_s$ constant is strongly dependent on the characteristics of the semiconductor used. In the implemented prototype, transition times less than 3.5 ms were obtained by adjusting the slope of the ramp-circuit control of $v_{GE}$ as shown in Figure 10b.

Figure 10 summarizes the nature of the hybrid control solution implemented in the prototype [30,31].

This type of control naturally leads to the appearance of IGBT S collector-emitter voltage ($v_{CE}$) voltage peaks that might decrease the time life of the semiconductors unless safety circuits are considered in the design of the power supply. The amplitude of the voltage peaks can be estimated using Equation (10) since $v_{GE} = v_s$. For a design specification of $\Delta t \sim 3$ ms, $v_{CE}$ might reach values over 500 V risking damaging the semiconductor.

In the implemented prototype, this problem is prevented with the use of a varistor and a snubber (see Figure 2) that makes it possible to keep the IGBT operation within its

"safe operating zone". Figure 11 illustrates the dynamic behavior of the $v_{CE}$ voltage and the typical amplitude and duration of the peaks for the developed power circuit. During the "upward" transition, the semiconductor is saturated and a voltage peak with a very short duration is observed when the dynamic transition ends due to a delay on the auxiliary switch ($S_{aux}$) turn off. During the "downward" transition, the $v_{CE}$ voltage steadies around a value established by the electromotive force in the magnet due to the decreasing current and in line with the control mode.

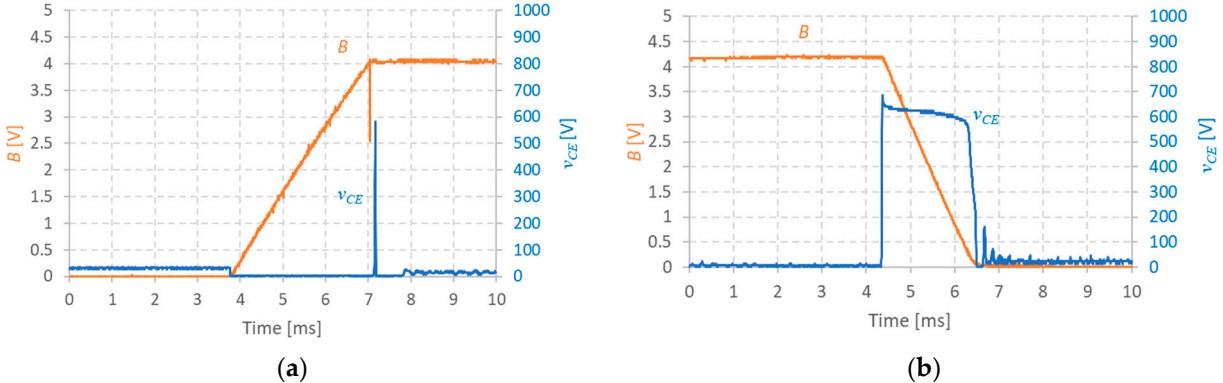

**Figure 11.** Experimental IGBT collector–emitter voltage for a magnetic field: (**a**) Upward transition; (**b**) Downward transition.

The implemented FFC-NMR power supply prototype can run continuously during several hours performing thousands of "upward" and "downward" transitions.

## 5. Conclusions

The design of new FFC-NMR power supplies for non-academic applications has been a continuous challenge [8,20,32,33]. Among the most important aspects are the volume size of the whole equipment, the electric power for operation and the cooling requirements.

In this work, we present a prototype designed to implement a low power solution with a single power semiconductor for a ferro-electromagnet. The power supply operates with a hybrid current control. Contrary to classical applications where IGBTS operate mostly as ON/OFF switches (saturation/cut-off), the hybrid control allows for a continuous cycling between the ON/OFF mode and linear control of the power semiconductor. This approach presents some technical novelties with potential advantages:

On one hand, the control includes a conventional PI controller at the core of the system that can be used to explore the particular characteristics of different types of semiconductors (IGBT and MOSFET) in a modular way. On the other hand, the semiconductors are controlled in non-conventional ON/OFF and linear alternating modes.

The transient times intrinsic to the FFC-NMR operation can be adjusted with the use of $U_{aux}$ for the upward transitions and the slew-rate of the ramp-circuit for the downward transition.

Contrary to other solutions, which require balanced and stable banks of transistors the prototype presented here requires a reduced number of power semiconductors and constitutes a portable low power solution for FFC-NMR.

As a consequence, the equipment is expected to require less maintenance and constitutes a step forward in the evolution of this type of system aiming at the design of more sustainable and efficient solutions.

Pictures of the power supply front panel and of the portable magnet are shown in Figures 12a and 12b, respectively.

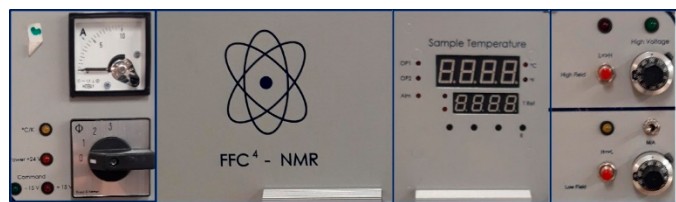
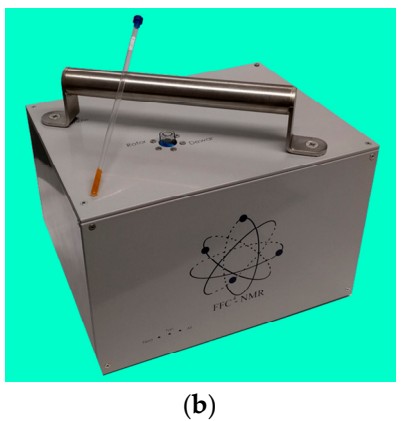

(**a**)　　　　　　　　　　　　　　　　　　　　　　(**b**)

**Figure 12.** Pictures of the experimental setup: (**a**) Power supply; (**b**) Magnet.

**Author Contributions:** Conceptualization, A.R., D.M.S., P.J.S. and E.M.; methodology, A.R., D.M.S. and P.J.S.; software, P.J.S.; validation, A.R., D.M.S., V.S. and P.J.S.; formal analysis, D.M.S. and P.J.S.; investigation, A.R., D.M.S., P.J.S. and V.S.; resources, P.J.S.; data curation, A.R., D.M.S. and P.J.S.; writing—original draft preparation, A.R., D.M.S. and P.J.S.; writing—review and editing, D.M.S. and P.J.S.; visualization, A.R., D.M.S. and P.J.S.; supervision, D.M.S. and P.J.S.; project administration, P.J.S.; funding acquisition, P.J.S. All authors have read and agreed to the published version of the manuscript.

**Funding:** This research received no external funding.

**Data Availability Statement:** Data will be available on request.

**Conflicts of Interest:** The authors declare no conflict of interest.

## Abbreviations

| | |
|---|---|
| FFC | Fast Field Cycling |
| IGBT | Insulated Gate Bipolar Transistor |
| MOSFET | Metal Oxide Semiconductor Field Effect Transistor |
| NMR | Nuclear Magnetic Resonance |
| PI | Proportional-Integral |

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
