# Peer review of "FFC-NMR Power Supply with Hybrid Control of the Semiconductor Devices"

_jlpea, doi:10.3390/jlpea13030052_

Round 1

Reviewer 1 Report

The manuscript titled "FFC-NMR Power Supply with Hybrid Control of the Semiconductors" introduces new hybrid solution to control power semiconductors as switches or as a linear device. The approach followed avoids the over-design of the power supply and make possible to implement new low power solutions. The manuscript needs major revisions to elaborate its novelty and other details in order to make more impacts.

Comment 1: The quality of figures needs to be improved. e.g. Figure 10b, remove the negative numbers in y axis;  Figure 11b, adjust the size to match with a.

Comment 2: The manuscript simply concludes that a prototype runs continuous cycling between ON/OFF mode. What is the significance of the work? The authors seem to expand in the Conclusion. However, this point needs to be emphasized throughout the manuscript so the readers can get more rapidly. Need to highlight at least in the Introduction-the motivation, novelty, significance and current research status.

Comment 3: More references are expected to indicate the research progress and supporting information of this area, especially publications after 2020 for current studies. Please cite 5-10 more if the journal requirement allows.

Comment 4: Figure 11 needs more description for the dynamic behaviors.

Author Response

The authors would like to thank the reviewer for their comments and suggestions. We are sure that these comments and suggestions led to a very substantial improvement on the paper, addressing the comments and suggestions of the reviewer:

  1. Figures 10 and 11 were changed.
  2. The authors appreciate and agree with this comment. In the abstract (lines17-18) was added “…constituting a novel design by joining a continuous match between the ON/OFF mode and the linear control of the power semiconductor devices.”. In section 1 was also written (lines 64-67) “Merging the non-conventional ON/OFF and linear control modes constitutes a technical novelty in the FFC-NMR field that led to a low power solution with a single power semiconductor for a ferro-electromagnet.”.
  3. The authors are in line and appreciate this comment. Anyway, the authors should refer that Editor suggested this submission as “communication” type and so that is limited in size and extent, and so that, the option was to concentrate te “communication” in the novelty aspects of the research done. Furthermore, the majority of publication in the last couple of years in this field (FFC NMR apparatus development) belong to the authors. Previously to the reviewing step, the authors agreed with the Editor to decrease the number of self-citations and to rearrange the references’ list as it is now.
  4. The authors agree and included in lines 206-211 the following explanation: “During the ”upward” transition the semiconductor is saturated and a voltage peak with a very short duration is observed when the dynamic transition ends due to a delay on the auxiliary switch (Saux) turn off. During the “downward” transition the vCE voltage steadies around a value established by the electromotive force in the magnet due to the decreasing current and in line with the control mode.”

Reviewer 2 Report

The authors may improve the paper along the following directions.

1.In the Section 2, the author's description of the parameter alpha needs to be more detailed.

2.Figure 6 needs to be improved, for example, Figure 6.b and Figure 6.a are not obvious in Figure 6.

3.The proportion of literature in the last five years needs to be increased.

good

Author Response

RESPONSE

The authors would like to thank the reviewer for their comments and suggestions. We are sure that these comments and suggestions led to a very substantial improvement on the paper, addressing the comments and suggestions of the reviewer:

  1. The authors thank and agree this comment, and so that had included in lines 83-86 : “Therefore, the parameter  must be set according to each type of transient and the characteristics of the power semiconductor. Technically, this parameter can be estimated from the technical vCE=f(iCE) curves provided by the manufacturers or obtained experimentally by testing the power semiconductor.”.
  2. Thank you for alerting. Correction done.
  3. The authors are in line and appreciate this comment. Anyway, the authors should refer that Editor suggested this submission as “communication” type and so that is limited in size and extent, and so that, the option was to concentrate te “communication” in the novelty aspects of the research done. Furthermore, the majority of publication in the last couple of years in this field (FFC NMR apparatus development) belong to the authors. Previously to the reviewing step, the authors agreed with the Editor to decrease the number of self-citations and to rearrange the references’ list as it is now.

Reviewer 3 Report

In the mansucript "FFC-NMR Power Supply with Hybrid Control of the Semiconductors", authors proposed a hybrid control approach on the operation of the power devices in the FFC-NMR power supply. There are some merits in the manuscript. However, the following issues should be addressed:

1. The title of the manuscript is not adequate. It is not clear on the term "the Semiconductors". Insteadly, it should be termed clearly, e.g. semiconductor devices or power devices.

2. The introduction part should be rewritten. The research background and the other publication results should be reviewed. 

3.There are some typos in the manuscript. For example, in the conclusion section, the first sentence "The design of new FFC-NMR power supplies for non-academic applications has been a continuous challenge [8], [20], [32-23]." ."[32-23]" should be corrected.

4. It is confused to read Fig.6. In the text, authors mentioned Fig.6(a) and Fig6(b). It should be supplemented.

5. The captions of each figures are too simple. Authors should supplement the detailed description in the captions.

6.For the magnet, authors mentioned a parameter "magnetic flux density/magnet current ratio B/im". However, in the following design, no consideration on the parameter. Supplementaion should be made on the consideration.

The overall English is acceptable. A few typos and grammar mistakes exist. 

Author Response

The authors would like to thank the reviewer for their comments and suggestions. We are sure that these comments and suggestions led to a very substantial improvement on the paper, being the English revised and addressing the comments and suggestions of the reviewer:

  1. The authors thank this comment, and changed the title to: “FFC-NMR Power Supply with Hybrid Control of the Semi-conductor devices”.
  2. The authors are in line and appreciate this comment. Anyway, the authors should refer that Editor suggested this submission as “communication” type and so that is limited in size and extent, and so that, the option was to concentrate te “communication” in the novelty aspects of the research done. Furthermore, the majority of publication in the last couple of years in this field (FFC NMR apparatus development) belong to the authors. Previously to the reviewing step, the authors agreed with the Editor to decrease the number of self-citations and to rearrange the references’ list as it is now.
  3. The authors thank this comment. This and other corrections were performed along the text.
  4. Thank you for alerting. Correction done.
  5. The authors appreciated this comment and changes were done in almost all the captions.
  6. The authors agree with this comment, and the following explanation was introduced in lines 160-164: “Using a calibrated Hall probe, it was observed experimentally, that for a Bref = 4.2 V corresponds a real B of 0.2T and a magnet current iM of 3.8 A. Clearly, the dynamic behavior of B, in Fig. 9, and iM, in Fig. 8, is very similar, from which a B/iM ratio of about 0.053 T/A is observed, in line with the ratio of 0.05T/A initially specified.”.

Round 2

Reviewer 1 Report

The manuscript has been revised accordingly with a clearer presentation and overall improved quality. It is acceptable for the audience of JLPEA as in the present form.

Reviewer 3 Report

It is happy to see the concerns are addressed.